# Quality and Sensory Profile of Durum Wheat Pasta Enriched with Carrot Waste Encapsulates

**DOI:** 10.3390/foods11081130

**Published:** 2022-04-14

**Authors:** Vanja Šeregelj, Dubravka Škrobot, Jovana Kojić, Lato Pezo, Olja Šovljanski, Vesna Tumbas Šaponjac, Jelena Vulić, Alyssa Hidalgo, Andrea Brandolini, Jasna Čanadanović-Brunet, Gordana Ćetković

**Affiliations:** 1Faculty of Technology Novi Sad, University of Novi Sad, Bulevar Cara Lazara 1, 21000 Novi Sad, Serbia; vanjaseregelj@tf.uns.ac.rs (V.Š.); vesnat@uns.ac.rs (V.T.Š.); jvulic@uns.ac.rs (J.V.); jasnab@uns.ac.rs (J.Č.-B.); gcetkovic@uns.ac.rs (G.Ć.); 2Institute of Food Technology, University of Novi Sad, Bulevar Cara Lazara 1, 21000 Novi Sad, Serbia; dubravka.skrobot@fins.uns.ac.rs (D.Š.); jovana.kojic@fins.uns.ac.rs (J.K.); 3Institute of General and Physical Chemistry, Studentski Trg 12, 11000 Belgrade, Serbia; latopezo@yahoo.co.uk; 4Department of Food, Environmental and Nutritional Sciences (DeFENS), Università Degli Studi di Milano, Via Celoria 2, 20133 Milan, Italy; alyssa.hidalgovidal@unimi.it; 5Council for Agricultural Research and Economics—Centre for Animal Production and Aquaculture (CREA-ZA), Viale Piacenza 29, 26900 Lodi, Italy; andrea.brandolini@crea.gov.it

**Keywords:** carotenoids, cooking quality, nutritional quality, freeze-drying, spray-drying, durum wheat pasta, functional food

## Abstract

Consumer knowledge about pasta quality differs around the world. Modern consumers are more sophisticated compared to past times, due to the availability of information on pasta types and quality. Therefore, this study investigated the nutritional, physical, textural, and morphological quality of durum wheat pasta enriched with carrot waste encapsulates (10 and 20% freeze-dried encapsulate (FDE) and 10 and 20% spray-dried encapsulate (SDE)), as well as determining consumer preferences for this type of product. Replacement of semolina with FDE and SDE contributed to changes in the pasta nutritional quality, which was reflected in the increased protein, fat, and ash content. Additionally, changes in cooking quality, color, and texture were within satisfactory limits. The uncooked pasta enriched with 10 and 20% SDE was characterized by a lighter yellow intensity with color saturation, as well as an imperceptible waxy appearance compared to the control and enriched pasta with 10 and 20% FDE. After cooking, the yellow color was more intense in all the enriched pasta samples which can be linked to the raw cereal which was significantly greater in the control in comparison to the FDE and SDE containing samples. Overall, carrot waste can be a promising material for the food industry to produce high-quality pasta.

## 1. Introduction

Pasta is a widely consumed cereal-based product all over the world. Pasta products are very popular due to their nutritional compositions, long shelf life, availability in the market, low cost, simplicity of preparation, and transportation [1]. Furthermore, about 15 million tons of pasta are produced annually with an expected increase in the range between 5 and 10% [2]. In summary, the role of pasta in the daily human diet can hardly be overestimated in any country [3]. Consumer knowledge about pasta quality differs around the world, but modern consumers are more sophisticated compared to the past, due to the availability of a large amount of information about the types and quality of pasta and all other diet-related characteristics [4]. It is worth remarking that consumer perception of pasta quality is very complex to understand, but the majority regard primary texture, color, health-improving effect, as well as price as the most important characteristics in making buying decisions [3]. Although all pasta manufacturers want to provide products based on consumers’ preferences, they need to work within a good production practice, food safety principles, as well as functional food-management [5]. Targeting the mentioned requirements for pasta manufacturers, Hidalgo et al. [6] suggested fortification of pasta products, indicating that a partial replacement of flour with ingredients rich in nutritional and functional substances could be a good approach for the creation of higher pasta quality. One of those ingredients could be carrot waste rich in bioactive compounds because of the health-improving characteristics of its lipophilic compounds, primarily carotenoids [7,8,9]. Using carrot waste as a source rich in natural carotenoids provides an opportunity for fortification of food products, i.e., upgrading color properties and increasing antioxidant activity which is clinically related to several health benefits including inhibition of LDL oxidation, anti-inflammatory properties, alleviation of oxidative stress, and enhanced immune response [7]. On the other hand, using carotenoids in food products requires an encapsulation process due to the high sensitivity of these natural pigments to thermal degradation. The formation of a physical barrier for sensitive compounds provides longer shelf life under the variable storage conditions, preventing deleterious reactions and controlled release of targeted bioactives in food products [10,11]. Several scientific groups reported preliminary studies about pasta rich in carotenoids [12,13,14], but comprehensive research about this type of product and its quality and consumers’ preferences have not yet been reported. Due to this fact, the objectives of this study were to determine the nutritional, physical, textural, and morphological quality of durum wheat pasta enriched with carrot waste encapsulates, as well as to investigate the sensory profile for this type of functional food product.

## 2. Materials and Methods

### 2.1. Materials

After food processing in the beverage industry (“Nectar”, Bačka Palanka, Serbia), the obtained carrot waste was instantly sampled, freeze-dried, and kept at −20 °C until use. For carotenoid extraction, the sunflower oil (“Dijamant”, Zrenjanin, Serbia) was selected, while the whey protein concentrate, inulin, and durum wheat semolina were obtained from Olimp Laboratories (Debica, Nagawczyna, Poland), Elephant Pharma (Belgrade, Serbia), and Molino Pagani (Borghetto Lodigiano, Italy), respectively.

### 2.2. Carrot Waste Extraction and Encapsulate Preparation

Freeze-dried carrot waste was mixed with sunflower oil (1:10 *w*/*v*) at 25 °C by stirring and using time shifts of 10 min blend and 5 min pause to avoid heating. After centrifugation at 4000 rpm for 10 min, the supernatant was recovered and kept at refrigerator temperature protected from light. The obtained carrot waste oil extract was encapsulated by freeze-drying and spray drying techniques, according to the optimal conditions reported by Šeregelj et al. [7]. The optimum wall materials imply 100% whey protein for freeze-drying as well as 71% whey protein and 29% inulin for spray drying [7]. The first formulation was kept at −80 °C during 24 h and then freeze-dried at −40 °C for 48 h to ensure complete drying. The second formulation was spray dried at an inlet temperature of 130 °C and an outlet temperature of 65 ± 2 °C. The spray-dried encapsulates (SDE) and the freeze-dried encapsulates (FDE) were kept at −20 °C.

### 2.3. Pasta Manufacturing

Preparation of the pasta was carried out in a small-scale pilot plant (Mac30, Italpast, Parma, Italy) using the procedure described in detail in the author’s previous work [9]. Briefly, the control pasta dough was produced from durum wheat semolina (32% final humidity), while the carrot waste enriched pastas were created by replacing semolina with 10% or 20% FDE or SDE (Figure 1).

### 2.4. Nutritional Quality

The pasta samples were examined for moisture (M), crude protein (CP), crude fat (CF), and ash according to the method described in AACC (2000), while total carbohydrate content (TC) was calculated by subtracting the sum of M, CP, CF, and ash from 100.

### 2.5. Cooking Quality

To assess pasta cooking quality, short-cut macaroni samples (100 g) were cooked in boiling water with salt addition (5 g/L). All samples were cooked for the optimum cooking time which was defined by squashing a cut-open macaroni between two glass plates at different cooking times. The pasta was considered cooked when the white, opaque core had disappeared.

#### 2.5.1. Cooked Weight

The cooked weight (CW) was measured as the weight of 100 g dry pasta after cooking.

#### 2.5.2. Swelling Index

The swelling weight index (SI) was determined based on the weight of cooked pasta (Wcp) dried to a constant mass (Wdp) at 105 °C [1]. It was calculated by using the Equation (1).
(1)$$\mathrm{SI}\;=\frac{\mathrm{Wcp}\;-\mathrm{Wdp}\;}{\mathrm{Wdp}}$$

#### 2.5.3. Protein Loss

Protein loss (PL) was determined as the amount of proteins measured in the cooking water and expressed as the percentage of total protein in the pasta.

### 2.6. Color Properties of Pasta

The Yellow Index (YIAE; YICE), the Brown Index (BIAE; BICE), and total color difference (∆E—DEAE; DECE) between uncooked and cooked pasta were used to evaluate pasta color properties. Color characteristics were measured by using the Minolta Chromameter (Model CR-400, Minolta Co., Osaka, Japan) equipped with attachment CR-A33b. All the samples were illuminated with D65-artificial daylight (10° standard angle). Each measure was performed in triplicate. YI [15], BI, and ∆E [16] were calculated following Equations (2)–(4):(2)$$\mathrm{YIAE}/\mathrm{YICE}\;=\frac{142.86\;\cdot {\;b}^{*}}{L^{*}}$$
(3)$$\mathrm{BIAE}/\mathrm{BICE}\;=100-{\;L}^{*}$$
(4)$$\Delta E^{*}=\sqrt{\Delta a^{*2}+\Delta b^{*2}+\Delta L^{*2}}$$
where L* is lightness, a* is red/green, and b* is yellow/blue. For evaluation of encapsulate addition (DEAE) the effect on ΔE, ΔL*, Δa*, and Δb* was calculated as the difference in L*, a*, and b* values (respectively) between control and enriched pasta samples. For evaluation of pasta cooking the effect on ΔE, ΔL*, Δa*, and Δb* was calculated as the difference in L*, a* and b* values (respectively) between uncooked and cooked pasta samples.

### 2.7. Textural Properties of Pasta

Optimally cooked pasta was washed with 500 mL of distilled water, drained, and allowed to balance at room temperature for 10 min in plates with lids before analysis. The texture analyzer (TA.XT Plus, Exponent Stable Micro System, Godalming, Surry, UK) was equipped with a 30 kg load cell, a P/36R probe was attached to the load cell, and the heavy duty platform was positioned centrally below the probe. Samples were centrally aligned under the retaining plate in as flat a position as possible. Pasta hardness (Hard), cohesiveness (Coh), springiness (Spr), chewiness (Chew), gumminess (Gum), and resilience (Res) were determined from the recorded force–time curve. The experimental procedure was as follows: 1 mm/s pre-test speed, 5 mm/s test speed and post-test speed, 75% strain, trigger type 5 g—auto. All texture measurements were carried out in six replicates.

### 2.8. Structural Morphology

Uncooked and cooked pastas were cut transversely without damaging the structure. The inner parts of the uncooked and cooked samples were used for analysis on a SEM Hitachi TM3030 scanning electron microscope (acceleration voltage 15 kV, beam current 20 nA, spot size 1 mm). For covering samples with gold the LeyboldHeraus L560Q putter coating device was used. The cooked pasta samples were dehydrated and prepared for SEM analysis using the preparation protocol reported by Ribotta et al. [17].

### 2.9. Sensory Analysis

The sensory profiles of uncooked and cooked pasta were determined by a trained sensory panel (2 males and 8 females, 23 to 45 years old) that consisted of members of the scientific team of the Institute of Food Technology, University of Novi Sad. A descriptive analysis was performed to obtain the complete description of a product’s sensory properties by using the checklist method for the selection of sensory descriptors [18]. The final list of sensory descriptors, reached after discussion and training sessions, is presented in Table 1. The perceived intensity of evaluated sensory properties was expressed on a 100 mm linear scale. The sensory analysis was performed in individual booths at 22 °C. The samples were presented in plastic closed boxes coded with three-digit random numbers and were evaluated in two consecutive sessions, within ten minutes after pasta cooking. The sample presentation was in a completely balanced order. Before sensory testing, all participants were asked about possible food allergies and were required to sign written consent to participate in the study.

### 2.10. Principal Component Analysis (PCA)

Principal component analysis (PCA) enables insight into the presence of patterns in available data by providing information of defined variables, which behave similarly to each other. The results of PCA analysis of the five samples according to the investigated variable nutritional characteristics (M, CP, CF, Ash, TC), cooking quality and color parameters (CW, SI, PL, DEAE, YIAE, BIAE, DECE, YICE, BICE), textural parameters (Hard, Spr, Coh, Gumm, Chew, and Res), and sensor profile (YIUP, CSUP, WAUP, YICP, COCP, FACP, BECP, FCP, SSCP, ETCP, Brit, Salt, FFI, Oil) were presented in the form of five biplot plots.

### 2.11. Standard Scores Analysis

For a more complex ranking investigation of durum wheat pasta enriched with carrot waste encapsulates, standard scores (SS) were evaluated by integrating the obtained values of different nutritional and cooking quality parameter evaluation methods. The min–max normalization was used to compare nutritional and cooking quality parameters of samples obtained using experiments, in which samples were ranked according to extreme values of experimental data. Normal standard scores of all variables, for each sample, were derived by the Equations (5) and (6).
(5)$${\bar{x}}_{i}=\frac{x_{i}\underset{i}{-\min}\;x_{i}}{\underset{i}{\max}\;x_{i}-\underset{i}{\min}\;x_{i}},\;\forall i,\;\mathrm{in}\;\mathrm{case}\;\mathrm{of}\;“\mathrm{the}\;\mathrm{higher},\;\mathrm{the}\;\mathrm{better}”\;\mathrm{criteria},\;\mathrm{or}$$
(6)$${\bar{x}}_{i}=1-\frac{x_{i}\underset{i}{-\min}\;x_{i}}{\underset{i}{\max}\;x_{i}-\underset{i}{\min}\;x_{i}},\;\forall i,\;\mathrm{in}\;\mathrm{case}\;\mathrm{of}\;“\mathrm{the}\;\mathrm{lower},\;\mathrm{the}\;\mathrm{better}”\;\mathrm{criteria},$$
where $x_{i}$ are the experimental data. The averaged normalized scores sum applied for each sample gives a unitless value, which is termed as “standard score” (SS).

### 2.12. Statistical Analysis

To assess differences for the traits analyzed among samples, one-way analyses of variance (ANOVA) were calculated. In the case where significant differences were discovered, Fisher’s least significant differences (LSD) at *p* ≤ 0.05 were determined. The data were enumerated statistically applying the results obtained in the software package XLSTAT July 2018.

## 3. Results

### 3.1. Nutritional Quality

Table 2 shows the nutritional composition of control and enriched durum wheat pasta with carrot waste encapsulates. The carrot waste encapsulate enrichment decreased moisture content and total carbohydrates, whereas it increased crude protein, crude fat, and ash. According to Gupta et al. [1], a greater protein–polysaccharides interaction in enriched samples compared to control leads to a reduction in moisture content. Significantly superior protein contents were detected in pasta enriched with freeze-dried carrot waste encapsulates (FDE) which was expected due to the content of whey protein in the wall material, while spray-dried encapsulates (SDE) included inulin (29%) as well in wall material. The increases in protein content were 3.12 and 5.37 g/100 g for the replacement of semolina with 10% and 20% of FDE. When semolina was replaced with 10% and 20% of SDE, the increases in protein content were 1.69 and 3.85 g/100 g. Crude fat contributed to approximately 0.8% of the control pasta weight. Enriched pasta with FDE and SDE showed a high crude fat content, ranging from 4.23 to 7.20 g/100 g. This increase could be due to the inclusion of nutrients such as fatty acids present in the encapsulated carrot waste oil extract [7]. The significantly lower carbohydrate content in enriched pasta samples could be attributed to the decrease in semolina level in the blend.

The PCA of the nutritional composition of durum wheat pasta samples showed that the first two principal components summarized 93.67% of the total variance in the five nutritional parameters (M, CP, CF, Ash, TC). According to the biplot of the PCA analysis of the collected data, moisture content (which provided 15.7% of the whole variance, established on correlations) and total carbohydrates (23.8%) showed a positive influence on the first principal component (PC1), while crude protein content (24.0%), crude fat content (20.6%) and ash content (16.0%) exerted a negative score in line with the PC1 component (Figure 2).

A positive leverage on the second principal component (PC2) was observed for moisture content (31.4% of the whole variance, identified on correlations) and ash content (50.3%), whereas a negative influence on PC2 was obtained for crude fat (13.1%). PC1 explained the differences in samples according to the nutritional composition of durum wheat pasta enriched with carrot waste encapsulates. Samples 20% FDE and 20% SDE achieved the required ash, crude protein, and crude fat content, while the control sample achieved the moisture and total carbohydrate content. The contents of ash and crude protein were higher in samples 10% FDE and 20% FDE.

### 3.2. Cooking Quality

Cooking pasta quality could be estimated based on cooked weight, swelling index, and protein loss. The results obtained for the cooking quality of pasta enriched with carrot waste encapsulates are shown in Table 3. Several authors have reported that the good cooking quality of pasta is related to the quality and content of protein, and the possibility to form optimum carbohydrates—protein network [2,19,20]. Table 3 illustrates that the replacement of semolina with carrot waste encapsulates pointedly affected the cooking weight of pasta; the values increased from 229.8 to 236.3 g and 237.5 g for pasta containing 10% and 20% of FDE respectively, while higher weights of 242.7 and 248.9 g were noted for pasta containing 10% and 20% of SDE respectively. The whey protein is rich in polar amino acids, so the increase in the cooked weight of pasta could be attributed to the water-binding capacity of the protein. On the other hand, SDE contains inulin which is highly hydrophilic and by this means the pasta cooking weight values increased more. Gupta et al. [1] reported cooking weight increase in quinoa protein isolate supplemented pasta because of the presence of polar amino acids. Reddy Surasani et al. [21] also found that the increased cooking weight of pasta with pangas protein isolate was a consequence of the protein being rich in polar amino acids.

Pasta enriched with carrot waste encapsulates showed a significantly higher swelling index than the control pasta (Table 3). The swelling index of the control was 0.77 g/g which increased in the range from 0.94 to 1.31 g/g when FDE or SDE was added and showed a directly proportional relation with increased concentration. Desai et al. [2] and El-Sohaimy et al. [22] obtained results which are in agreement with the present study. This property could be interpreted as referring to the water-binding and gelling ability of the proteins. Protein loss was not significantly affected by the substitution of semolina with 10% SDE, and thereafter it significantly increased for other enriched pasta. Higher values of protein loss were noted for pasta enriched with FDE, which could be due to 100% of whey protein in encapsulates and the high solubility of this protein. Mahmoud et al. [23] and Gupta et al. [1] also noticed an increase in protein loss for pasta with quinoa protein isolate and noodles fortified with protein products from lupine.

The PCA of the cooking quality of durum wheat pasta samples (Figure 3) showed that the first two principal components summarized 97.00% of the whole variance in the three parameters (CW, SI and PL). The cooked weight (which provided 19.3% of the whole variance, calculated based on correlations), swelling index (45.9%), and protein loss (34.8%) showed negative influence on PC1 (Figure 3). CW (70.0%) exerted a negative score according to PC2 component, while PL (30.0%) showed a positive influence on PC2 component. PL, SI, and CW parameters were the highest in samples 10% FDE and 10% FDE.

### 3.3. Color Properties

Color is the single most important food-intrinsic sensory cue and hence contributes to consumers differing expectations regarding the likely taste and flavor. In order to evaluate the pasta color change due to carrot waste encapsulate addition, the color differential index (ΔE) was defined. Table 3 illustrates that the ΔE (DEAE) values of carrot waste encapsulate enriched pasta increased with increasing levels of FDE and SDE, whereas significantly higher ΔE was exhibited by the SDE-containing pasta. For all samples, the DEAE values were more than 3.0, which means that color changes are perceptible by visual observation [24].

In durum wheat, lutein and zeaxanthin as representatives of carotenoids, are the main color components [9] and contribute substantially to the YIAE of semolina and pasta. The durum wheat pasta enriched with FDE is characterized by a significantly higher YIAE, which is a consequence of the presence of α-carotene, β-carotene, and cis β-carotene in encapsulated carrot waste extract [8]. On the other hand, the YIAE of durum wheat pasta enriched with SDE is lower than the control, and this could be explained by the presence of inulin in the composition of these samples. Additionally, the BIAE was significantly lower than in enriched pasta samples compared to the control. Giannone et al. [25] reported that a yellowish color and high YIAE are appreciated by consumers of durum wheat pasta, while BIAE should be low, allowing a perception of brilliant and luminous color in the final product. Cooked samples exhibited higher ΔE (DECE) values, indicative of the pigments released after cooking the pasta. This is also confirmed with higher YICE values and lower BICE values of cooked pasta samples. Gull et al. [26] also observed an increase in yellow color in cooked pasta from millet flour and carrot pomace and explained this change as the consequence of swelling and conversion of pigments during cooking.

The PCA of the color properties of durum wheat pasta samples showed that the first two principal components totaled 95.53% of the whole variance in the six parameters (DEAE, YIAE, BIAE, DECE, YICE, and BICE). BIAE (which contributed 24.0% of the total variance, calculated according to correlations), YIAE (16.0%), and DECE (21.9%) exhibited a positive influence on PC1, while YICE (22.3%) and DEAE (15.5%) exerted a negative score according to the PC1 component. The positive influence on PC2 was exerted by BICE (56.5% of the total variance, based on correlations), while the negative influence on PC2 was obtained for DEAE (20.5%) and YIAE (16.6%) (Figure 4).

### 3.4. Textural Properties

The texture of pasta is one of the most significant indicators of quality that influences sensory attributes and final consumer acceptance. Table 3 illustrates the textural properties of cooked pasta represented by hardness, springiness, cohesiveness, gumminess, chewiness, and resilience parameters. An increase in hardness was recorded when comparing control and enriched pasta with carrot waste encapsulates. Ogawa and Adachi [27] ascribed the strength of the gluten network as the main factor which governed the hardness of enriched pasta. The samples enriched with 20% FDE and SDE showed a significant decrease in springiness and cohesiveness. This meant that for the 20% enriched pasta, it was more difficult to hold the structure together as time proceeded [28]. Gumminess and chewiness showed increasing values for the increasing concentration of FDE in pasta, while in terms of pasta enriched with SDE these values were lower. It was also observed that the addition of carrot waste encapsulates up to 20% did not have a significant influence on pasta resilience.

The PCA of the textural properties of durum wheat pasta samples (Figure 5) showed that the first two principal components totaled 54.96% of the whole variance in the six textural properties (hardness, springiness, cohesiveness, gumminess, chewiness, and resilience). Springiness (which contributed 16.4% of the total variance, based on correlations), cohesiveness (10.1%), gumminess (17.8), chewiness (26.4%), and resilience (24.4%) exhibited negative influence on PC1 (Figure 2). The positive influence on the second principal component (PC2) was noticed for cohesiveness (23.4% of the total variance, based on correlations), and springiness (16.2%), whereas a negative influence on PC2 was obtained for gumminess (16.6%) and hardness (32.3%). The textural properties were augmented in the samples of control, 10% FDE, and 10% SDE.

### 3.5. Standard Score

The “higher the better” or the “lower the better” criteria were used according to the sign in the “Polarity” column in Table 2 and Table 3 for nutritional quality and cooking quality, color, and textural properties of durum wheat pasta enriched with carrot waste encapsulates. The standard score (SS) was calculated using Equation (7) by summing the normal scores for all variables which were multiplied by their weight.
(7)$$\mathrm{SS}\;=\frac{\bar{X_{1}}+\bar{X_{2}}+\ldots +\bar{X_{n}}}{n}$$
where $\bar{X_{i}}$ were the nutritional quality and cooking quality, color, and textural properties defined in Table 2 and Table 3. The maximum of SS represents the optimal nutritional quality and cooking quality, color, and textural parameters. SS evaluation results are presented in Figure 6. The optimal parameters were for the sample 20% SDE, which exhibited the highest SS value of 0.662.

### 3.6. SEM Observations

On the micrographs (Figure 7), the surfaces of the pasta are present in dry form (uncooked) and after contact with water during the whole cooking process (cooked). Differences in structure can be observed depending on the different stage of gelatinization, the fibrillar structure of the protein network, or some of its fragments. Regardless of the type of pasta sample, uncooked pasta samples have an irregular outer surface in which starch granules are entirely entrenched in the gluten matrix. It is well-known that gluten exists with irregular edges and starch granules are distributed in the direction of the force applied during the initial phase of extrusion [29]. Considering that all the tested pasta samples were made in the same extruder, the pattern in the gluten network was similar, which was to be expected. The micrographs of the samples showed a loose fibrillar protein network which was subject to big changes after the cooking process. This is in agreement with reported remarks by Gull et al. [26], who explained that the cooking process expands the pasta in volume resulting in the enveloping protein film and pasta surface becoming smoother. These complex changes are reflected in the appearance of cavities of different sizes and shapes (origin from gluten), as well as starch granules in different phases of gelatinization. Comparing the micrographs obtained for 10 and 20% SDE pasta, as well as 10 and 20% FDE pasta, differences in the structure of the protein network and the degree of deformation of the starch granules can be observed. Briefly, for these samples only the fibrillar structure of the protein network is visible, and the starch granules are completely gelatinized or deformed and decomposed. The difference in rupture surface is more evident compared to the surface micrographs of enriched pasta with control samples. Observed morphological differences between control and enriched pasta samples can be related to differences in the quality characteristics of the examined pasta (texture, behavior during cooking, sensory quality, etc.).

### 3.7. Sensory Analysis

Descriptive sensory values are summarized in Figure 8. The uncooked pasta samples enriched with 10 and 20% SDE were characterized by lighter yellow intensity and color saturation, as well as imperceptible waxy appearance compared with the control and enriched pasta with 10 and 20% FDE. Furthermore, uncooked 20% FDE pasta was marked with the highest values of yellow color intensity and saturation, followed by 10% FDE and the control pasta. These results are comparable with the previously presented comments for the color properties in Section 3.3. The very pronounced waxy appearance of 20% FDA pasta is probably because of the high carrot waste extracted in oil delivered by encapsulates [7,9]. On the other hand, after cooking, the yellow color was more intensive in all the enriched pasta samples, which is in good agreement with the findings reported by Gull et al. [26]. The intensity of odour linked with raw cereal was significantly more intense in the control in comparison to the FDE and SDE containing samples. Namely, the addition of FDE and SDE in pasta contributed to a more intense fat and boiled egg odour. Therefore, the more intensive flavour associated with fat or oil was thought to be in these samples, due to the presence of sunflower oil in the encapsulate composition [9]. The saltiness, i.e., the salty taste associated with sodium chloride solution was barely noticeable in enriched pasta samples, while in the control this taste was more intensive. Carrot waste encapsulate addition meaningfully contributed to changes in the textural properties of the pasta. The firmness of cooked pasta reflects the force required to bite down on pasta strands between the molars. Enriched pasta with FDE and SDE was characterized as firmer than control.

The obtained results are in agreement with the report by Gupta et al. [1] which explained that the thermal protein denaturation during cooking enhances the firmness. Laleg et al. [30] also reported that protein additives increase the firmness of pasta. Since the protein content is higher in FDE pasta samples, higher values of firmness were recorded for these samples. The surface stickiness of FDE and SDE enriched pasta was lower than control, which may be attributed to the addition of whey protein and sunflower oil. The protein network entraps the starch granules which then prevent starch leaching [31], while sunflower oil acts as a lubricant. Elasticity was inversely dependent on the protein content. Pasta enriched with encapsulated carrot waste extract with pure whey protein was characterized as barely elastic and very brittle. The control pasta was almost found to have no oiliness, i.e., none of the oily sensations in the mouth remained after pasta swallowing. Samples enriched with carrot waste encapsulates were described with a noticeable more intense oil sensation after swelling, which is to be expected, given the composition of these samples.

The PCA of the sensory analysis of durum wheat pasta samples showed that the first two principal components totaled 92.11% of the total variance in the fourteen sensory analysis parameters. COCP (which contributed 10.0% of whole variance, calculated, according to correlations), ETCP (10.3%) and Salt (10.0%) exhibited a positive influence on PC1, while FCP (8.4%), YICP (9.5%), Oil (9.9%), FACP (10.2%), FFI (10.3%), and Brit (8.3%) showed a negative score according to PC1 component (Figure 9). The positive influence on PC2 was noticed for WAUP (28.2%), YIUP (28.5%), and CSUP (19.0%), while a negative influence on PC2 was obtained for BECP (10.3%).

PC1 explained the differences in the cooked pasta samples according to the sensory analysis parameters, while PC2 explained the differences in the uncooked durum wheat pasta samples enriched with carrot waste encapsulates (Figure 8). Samples 10% FDE and 20% FDE were characterized by augmented Firmness, Yellow color intensity, Oiliness, Fat odour intensity, Fat flavour intensity, Brittleness and Boiled egg odour intensity in cooked pasta, while the control sample was characterized by increased Cereal odour intensity, Elasticity tactile, Saltiness, and Surface stickiness in cooked pasta. The increased values of Waxy appearance, Yellow color intensity, and Color saturation were noticed in the 10% FDE uncooked pasta sample.

## 4. Conclusions

Pasta enrichment with carrot waste encapsulates significantly improved the protein, fat and ash contents. In addition, cooking quality, color, and textural properties were affected within acceptable limits. Sensory descriptive analysis revealed that enriched durum wheat pasta with 10% FDE exhibited more intensive yellow color intensity, color saturation, and waxy appearance. Enriched samples were also characterized by lower surface stickiness. Overall, the supplementation of durum wheat semolina with encapsulated carrot waste extract could be a good approach for producing pasta with better nutritional ingredients and satisfactory technical and sensory qualities.

## Figures and Tables

**Figure 1 foods-11-01130-f001:**
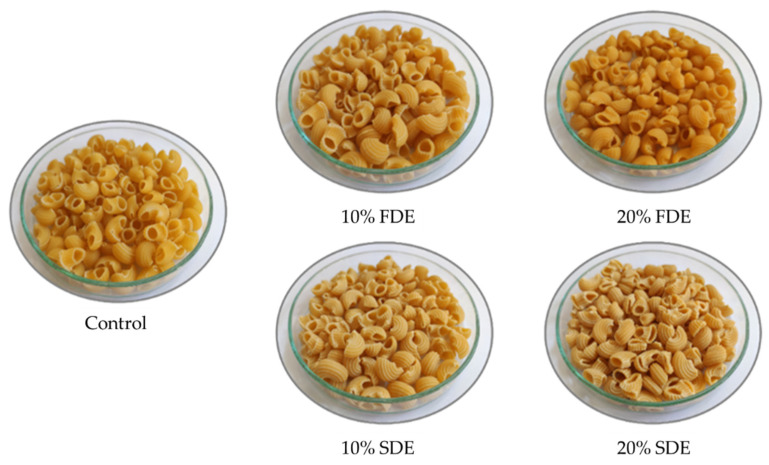
Manufactured pasta. Control: Durum wheat pasta without carrot waste encapsulates. 10% FDE: Durum wheat pasta with 10% freeze-dried carrot waste encapsulate. 10% SDE: Durum wheat pasta with 10% spray-dried carrot waste encapsulate. 20% FDE: Durum wheat pasta with 20% freeze-dried carrot waste encapsulate. 20% SDE: Durum wheat pasta with 20% spray-dried carrot waste encapsulate.

**Figure 2 foods-11-01130-f002:**
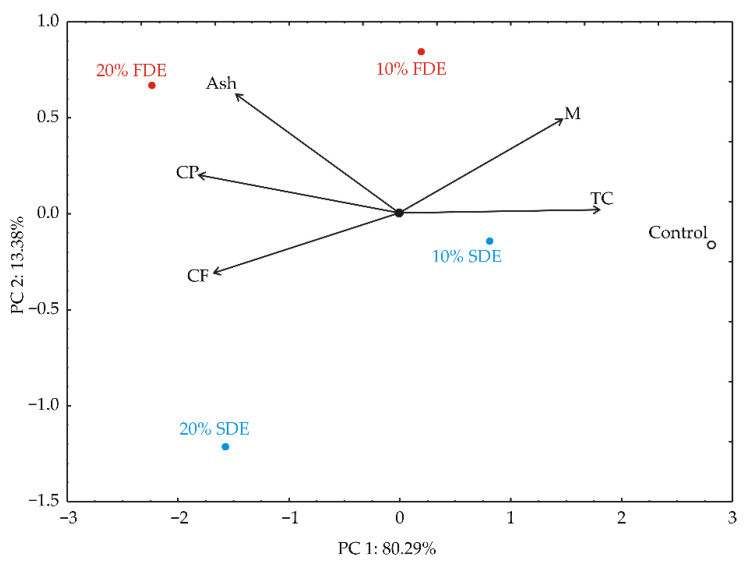
PCA ordination of variables based on the nutritional composition (M—moisture; CP—crude protein; CF—crude fat; TC—total carbohydrate content). Control: Durum wheat pasta without carrot waste encapsulates. 10% FDE: Durum wheat pasta with 10% freeze-dried carrot waste encapsulate. 10% SDE: Durum wheat pasta with 10% spray-dried carrot waste encapsulate. 20% FDE: Durum wheat pasta with 20% freeze-dried carrot waste encapsulate. 20% SDE: Durum wheat pasta with 20% spray-dried carrot waste encapsulate.

**Figure 3 foods-11-01130-f003:**
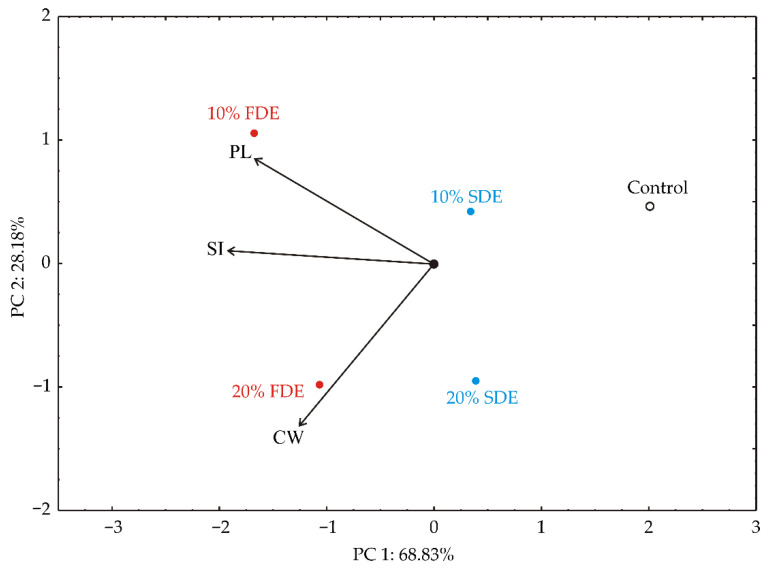
PCA ordination of variables based on cooking quality of durum wheat pasta enriched with carrot waste encapsulates (CW—Cooked weight (g); SI—Swelling Index (g/g); PL—Protein loss (%)). Control: Durum wheat pasta without carrot waste encapsulates. 10% FDE: Durum wheat pasta with 10% freeze-dried carrot waste encapsulate. 10% SDE: Durum wheat pasta with 10% spray-dried carrot waste encapsulate. 20% FDE: Durum wheat pasta with 20% freeze-dried carrot waste encapsulate. 20% SDE: Durum wheat pasta with 20% spray-dried carrot waste encapsulate.

**Figure 4 foods-11-01130-f004:**
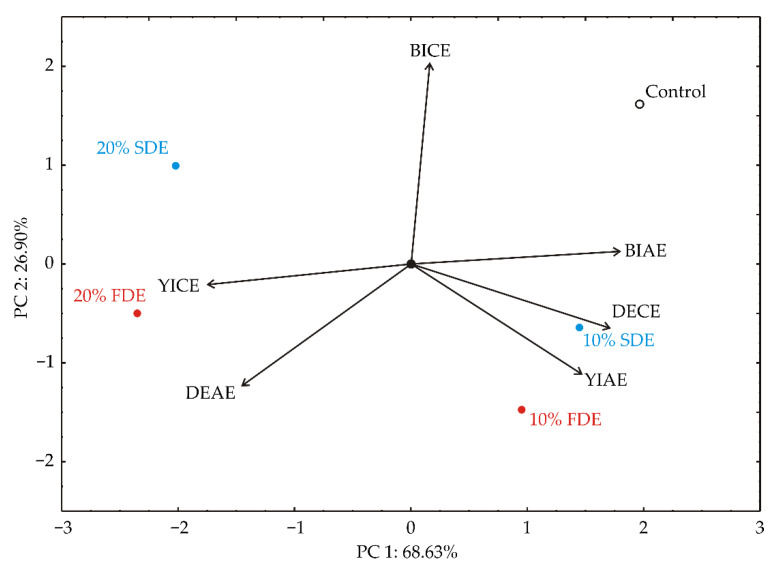
PCA ordination of variables based on color properties of durum wheat pasta enriched with carrot waste encapsulates (DEAE—ΔE encapsulate addition effect; YIAE—yellow index addition effect; BIAE—brown index addition effect; DECE—ΔE cooking effect; YICE—yellow index cooking effect; BICE—brown index cooking effect). Control: Durum wheat pasta without carrot waste encapsulates. 10% FDE: Durum wheat pasta with 10% freeze-dried carrot waste encapsulate. 10% SDE: Durum wheat pasta with 10% spray-dried carrot waste encapsulate. 20% FDE: Durum wheat pasta with 20% freeze-dried carrot waste encapsulate. 20% SDE: Durum wheat pasta with 20% spray-dried carrot waste encapsulate.

**Figure 5 foods-11-01130-f005:**
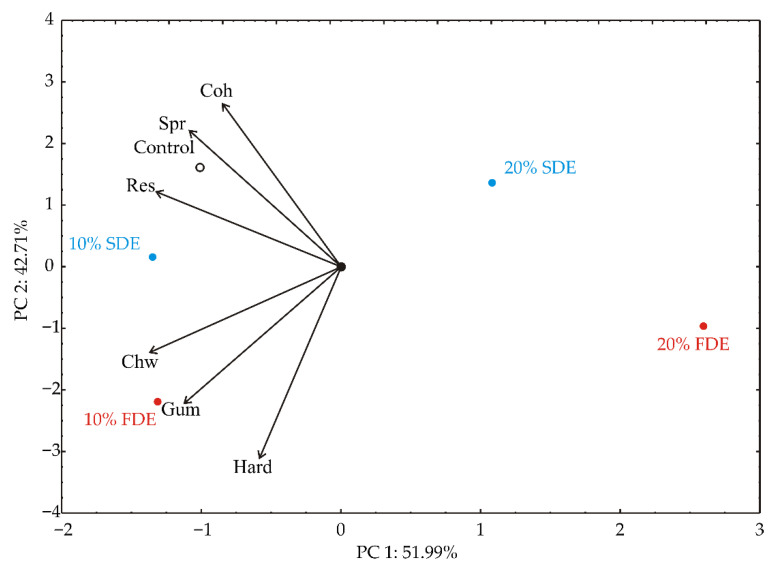
PCA biplot of variables based on textural properties of durum wheat pasta enriched with carrot waste encapsulates (Hard—Hardness (N); Spr—Springiness; Coh—Cohesiveness; Gum—Gumminess; Chw—Chewiness; Res—Resilience). Control: Durum wheat pasta without carrot waste encapsulates. 10% FDE: Durum wheat pasta with 10% freeze-dried carrot waste encapsulate. 10% SDE: Durum wheat pasta with 10% spray-dried carrot waste encapsulate. 20% FDE: Durum wheat pasta with 20% freeze-dried carrot waste encapsulate. 20% SDE: Durum wheat pasta with 20% spray-dried carrot waste encapsulate.

**Figure 6 foods-11-01130-f006:**
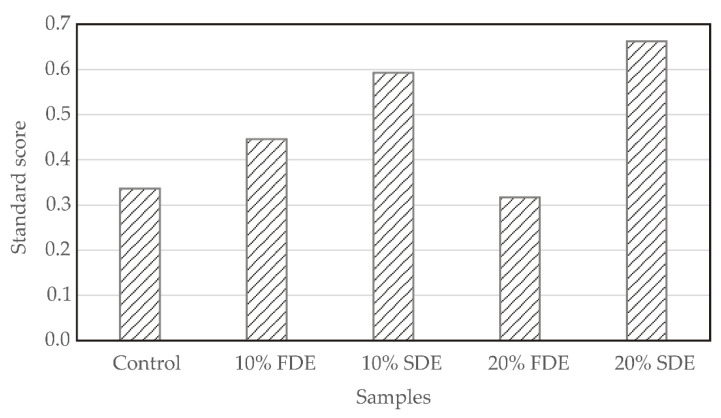
Standard score for samples. Control: Durum wheat pasta without carrot waste encapsulates. 10% FDE: Durum wheat pasta with 10% freeze-dried carrot waste encapsulate. 10% SDE: Durum wheat pasta with 10% spray-dried carrot waste encapsulate. 20% FDE: Durum wheat pasta with 20% freeze-dried carrot waste encapsulate. 20% SDE: Durum wheat pasta with 20% spray-dried carrot waste encapsulate.

**Figure 7 foods-11-01130-f007:**
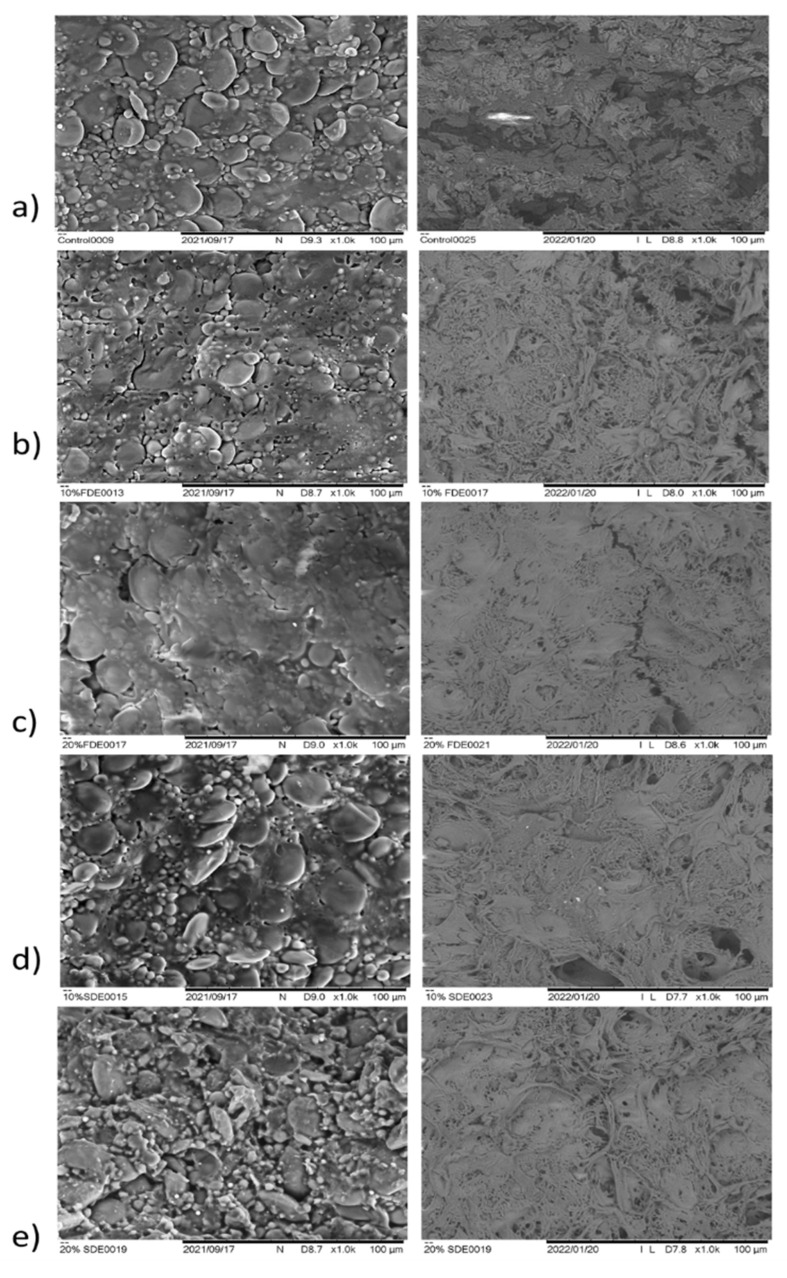
SEM images of uncooked and cooked pasta: (**a**) Control: Durum wheat pasta without carrot waste encapsulates. (**b**) 10% FDE: Durum wheat pasta with 10% freeze-dried carrot waste encapsulate. (**c**) 20% FDE: Durum wheat pasta with 20% freeze-dried carrot waste encapsulate. (**d**) 10% SDE: Durum wheat pasta with 10% spray-dried carrot waste encapsulate. (**e**) 20% SDE: Durum wheat pasta with 20% spray-dried carrot waste encapsulate.

**Figure 8 foods-11-01130-f008:**
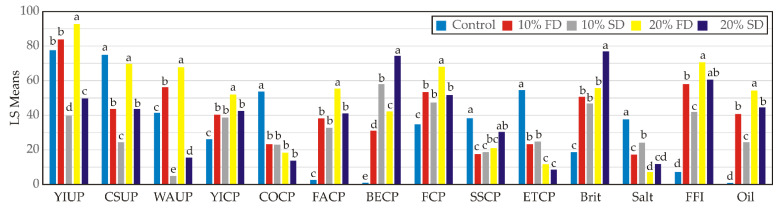
Descriptive sensory analysis of of durum wheat pasta enriched with carrot waste encapsulates. YIUP—Yellow color intensity (uncooked pasta); CSUP—Color saturation (uncooked pasta); WAUP—Waxy appearance (uncooked pasta); YICP—Yellow color intensity (cooked pasta); COCP—Cereal odour intensity (cooked pasta); FACP—Fat odour intensity (cooked pasta); BECP—Boiled egg odour intensity (cooked pasta); FCP—Firmness (cooked pasta); SSCP—Surface stickiness (cooked pasta); ETCP—Elasticity tactile (cooked pasta); Brit—Brittleness; Salt—Saltiness (cooked pasta); FFI—Fat flavour intensity (cooked pasta); Oil—Oiliness. Control: Durum wheat pasta without carrot waste encapsulates. Pasta samples with: 10% FD: 10% freeze-dried carrot waste encapsulate. 20% FD: 20% freeze-dried carrot waste encapsulate. 10% SD: 10% spray-dried carrot waste encapsulate. 20% SD: 20% spray-dried carrot waste encapsulate. Different letters in superscripts within the same row are significantly different at *p* ≤ 0.05 according to Fisher’s least significant differences (LSD) test.

**Figure 9 foods-11-01130-f009:**
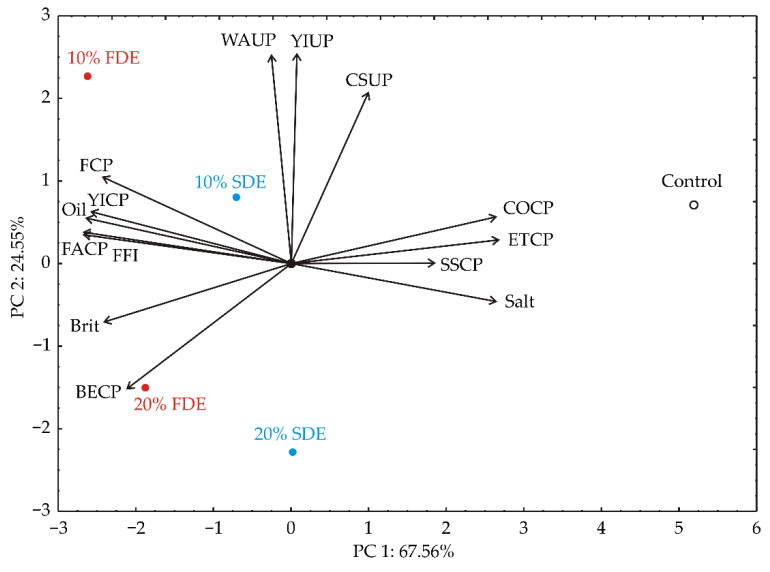
PCA ordination of variables based on sensory analysis of durum wheat pasta enriched with carrot waste encapsulates. YIUP—Yellow color intensity (uncooked pasta); CSUP—Color saturation (uncooked pasta); WAUP—Waxy appearance (uncooked pasta); YICP—Yellow color intensity (cooked pasta); COCP—Cereal odour intensity (cooked pasta); FACP—Fat odour intensity (cooked pasta); BECP—Boiled egg odour intensity (cooked pasta); FCP—Firmness (cooked pasta); SSCP—Surface stickiness (cooked pasta); ETCP—Elasticity tactile (cooked pasta); Brit—Brittleness; Salt—Saltiness (cooked pasta); FFI—Fat flavour intensity (cooked pasta); Oil—Oiliness.

**Table 1 foods-11-01130-t001:** Sensory attributes, descriptors, and definitions with end anchors.

Sensory Attributes	Descriptors	Definition with End Anchors
Appearance	Yellow color intensity—YIUP (uncooked pasta)	The intensity of yellow color (light—dark)
	Color saturation—CSUP (uncooked pasta)	The degree of color pureness (relative to pure grey) (washed out/pale—pure/vivid)
	Waxy appearance—WAUP (uncooked pasta)	Resembling wax in appearance (imperceptible—very pronounced)
	Yellow color intensity—YICP (cooked pasta)	The intensity of yellow color (light—dark)
Odour	Cereal odour intensity—COCP (cooked pasta)	The intensity of odour associated with raw cereals topped with boiling water (none—intensive)
	Fat odour intensity—FACP (cooked pasta)	The intensity of odour associated with fat or oil (none—intensive)
	Boiled eggs odour intensity—BECP (cooked pasta)	The intensity of odour associated with boiled eggs (none—intensive)
Flavour	Fat flavour intensity—FFI (cooked pasta)	The intensity of flavour associated with fat or oil (none—intensive)
Taste	Saltiness—Salt (cooked pasta)	The intensity of salty taste associated with sodium chloride solution (none—intensive)
Texture	Firmness—FCP (cooked pasta)	Force required biting down on pasta strands between the molars (not at all firm—very firm)
	Surface stickiness—SSCP (cooked pasta)	The degree to which pasta strands adhering to each other (not at all sticky—very sticky)
	Elasticity tactile—ETCP (cooked pasta)	Ability of the sample to return to the starting position after compression (not at all elastic—very elastic)
	Brittleness—Brit	The tendency of pasta to break without being significantly exposed to a high level of stress (not at all brittle—very brittle)
Residual	Oiliness—Oil	The degree to which the oily sensation in the mouth lags behind after pasta swallowing (none—intensive)

**Table 2 foods-11-01130-t002:** Nutritional quality of pasta enriched with carrot waste encapsulates.

Composition (g/100 g)	Polarity	Control	10% FDE	10% SDE	20% FDE	20% SDE
Moisture (M)	−	9.63 ± 0.09 ^a^	9.61 ± 0.08 ^a^	9.60 ± 0.08 ^a^	8.87 ± 0.04 ^b^	8.73 ± 0.06 ^c^
Crude Protein (CP)	+	13.84 ± 0.04 ^e^	16.96 ± 0.09 ^c^	15.53 ± 0.06 ^d^	19.21 ± 0.12 ^a^	17.69 ± 0.08 ^b^
Crude Fat (CF)	+	0.79 ± 0.05 ^d^	4.23 ± 0.08 ^c^	4.32 ± 0.11 ^c^	5.51 ± 0.12 ^b^	7.20 ± 0.10 ^a^
Ash	−	0.84 ± 0.02 ^c^	0.91 ± 0.02 ^b^	0.87 ± 0.00 ^bc^	0.97 ± 0.04 ^a^	0.88 ± 0.03 ^bc^
Total carbohydrates (TC)	−	74.90 ± 0.04 ^a^	68.29 ± 0.07 ^c^	69.68 ± 0.06 ^b^	65.40 ± 0.08 ^d^	65.50 ± 0.16 ^d^

Results are presented in form of mean ± standard deviation (*n* = 3). Different letters in superscripts within the same row are significantly different at *p* ≤ 0.05 according to Fisher’s least significant differences (LSD) test; Polarity: ‘+’ = the higher the better criteria, ‘−’ = the lower the better criteria.

**Table 3 foods-11-01130-t003:** Cooking quality, color, and textural properties.

	Polarity	Control	10% FDE	10% SDE	20% FDE	20% SDE
Cooking quality						
Cooked weight—CW (g)	+	229.8 ± 0.0 ^e^	236.3 ± 0.1 ^d^	242.7 ± 0.1 ^b^	237.5 ± 0.1 ^c^	248.9 ± 0.1 ^a^
Swelling Index—SI (g/g)	+	0.77 ± 0.0 ^e^	0.94 ± 0.1 ^d^	1.02 ± 0.0 ^c^	1.31 ± 0.0 ^a^	1.13 ± 0.1 ^b^
Protein loss—PL (%)	−	0.14 ± 0.01 ^c^	0.15 ± 0.01 ^b^	0.14 ± 0.00 ^c^	0.16 ± 0.01 ^a^	0.15 ± 0.00 ^b^
Color properties						
ΔE encapsulate addition effect—DEAE	+		3.96 ± 1.2 ^b^	6.89 ± 3.3 ^a^	6.58 ± 2.0 ^a^	8.16 ± 2.3 ^a^
Uncooked pasta						
YIAE		92.5 ± 3.1 ^c^	96.6 ± 2.3 ^b^	75.6 ± 4.2 ^d^	107.8 ± 1.5 ^a^	78.4 ± 4.4 ^d^
BIAE		43.0 ± 0.8 ^a^	41.6 ± 0.8 ^b^	34.8 ± 1.4 ^c^	40.5 ± 2.5 ^b^	32.8 ± 1.9 ^d^
Cooked pasta						
ΔE cooking effect—DECE	−	22.4 ± 2.1 ^b^	23.9 ± 1.4 ^ab^	11.9 ± 2.0 ^c^	24.9 ± 2.1 ^a^	12.0 ± 2.6 ^c^
YICE		45.9 ± 4.1 ^c^	46.1 ± 3.5 ^c^	57.5 ± 4.4 ^a^	51.7 ± 2.8 ^b^	57.2 ± 3.9 ^a^
BICE		25.1 ± 1.5 ^a^	23.2 ± 1.3 ^b^	24.7 ± 1.3 ^a^	23.2 ± 1.5 ^b^	23.2 ± 1.5 ^b^
Textural properties						
Hardness—Har (N)	−	16.0 ± 2.1 ^bc^	21.8 ± 2.1 ^b^	16.3 ± 3.5 ^c^	30.75 ± 7.6 ^a^	20.1 ± 1.6 ^b^
Springiness—Spr	+	0.99 ± 0.0 ^a^	0.96 ± 0.0 ^ab^	0.96 ± 0.1 ^ab^	0.90 ± 0.1 ^b^	0.77 ± 0.1 ^c^
Cohesiveness—Coh	−	0.66 ± 0.0 ^a^	0.54 ± 0.1 ^ab^	0.54 ± 0.0 ^ab^	0.48 ± 0.2 ^b^	0.42 ± 0.1 ^b^
Gumminess—Gum (N)	−	10.5 ± 1.7 ^b^	11.8 ± 2.5 ^ab^	7.24 ± 2.4 ^b^	14.9 ± 6.5 ^a^	8.81 ± 2.3 ^b^
Chewiness—Chew	+	10.4 ± 1.6 ^ab^	11.4 ± 2.6 ^ab^	6.94 ± 2.2 ^b^	13.3 ± 6.0 ^a^	6.80 ± 1.8 ^b^
Resilience—Res	−	16.0 ± 2.1 ^bc^	21.8 ± 2.1 ^b^	16.3 ± 3.5 ^c^	30.75 ± 7.6 ^a^	20.1 ± 1.6 ^b^

Results are presented in form of mean ± standard deviation (*n* = 3). Different letters in superscripts within the same row are significantly different at *p* ≤ 0.05 according to Fisher’s least significant differences (LSD) test. ΔE encapsulate addition effect: color difference between control and pasta with encapsulates. ΔE cooking effect: color difference between uncooked and cooked pasta. YIAE/YICE: yellow index. BIAE/BICE: brown index. Polarity: ‘+’ = the higher the better criteria, ‘−’ = the lower the better criteria.

## Data Availability

The data presented in this study are available in article.

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
