# Peer review of "Quality and Sensory Profile of Durum Wheat Pasta Enriched with Carrot Waste Encapsulates"

_foods, 2022, doi:10.3390/foods11081130_

Round 1
Reviewer 1 Report
The authors obtained carrot waste encapsulates and used them to enrich durum wheat pasta (macaroni). They studied the physicochemical, structural and textural characteristics of the pasta. The language is good; the experimental part is clear and the results are all very expected and logical. The manuscript has potential applications.
In the following lines, I give a detailed revision of the manuscript.
Best regards.
The abstract is very generic; the authors need to add the main results.
The sensory analysis was directed more towards descriptive analysis and not to sensory preference tests.
Line 83-85. The authors could add the concentrations of the encapsulating material with respect to the volume of waste oil extract.
Line 94. Why were these FDE or SDE values chosen to replace semolina in the paste?
Line 97. Crude fat or lipid content?
In the evaluation of optical properties (section 2.6), for total color difference (∆E*), which was the reference?
The significance letters generally reserve “letter a” for the largest mean in the tables.
Table 1. Why were moisture, protein and lipid contents higher when 20% BDS was used?
Please add the units for the textural properties reported in Table 3.
Line 333-345. The authors need to indicate the results from Table 3 in the discussion of texture.
Why did the authors not measure the carotenoid content in the carrot waste encapsulates and paste?
Author Response
The authors would like to thank the Editor and Reviewers for a professional review as well as the opportunity to make essential and crucial changes in our work. All Reviewers' remarks are accepted and the paper is changed according to comments. The authors believe that the changed paper would satisfy the Reviewer's criteria and that it is going to be acceptable for publishing in the Food.
We decided to revise the manuscript indicating with red letters all changes directly on the revised manuscript.
Best regards,
Dr. Olja Šovljanski and co-authors
Reviewer #1:
The authors obtained carrot waste encapsulates and used them to enrich durum wheat pasta (macaroni). They studied the physicochemical, structural, and textural characteristics of the pasta. The language is good; the experimental part is clear and the results are all very expected and logical. The manuscript has potential applications. In the following lines, I give a detailed revision of the manuscript.
Best regards.
AUTHORS: Thank you for all suggestions. We decided to take into account all your suggestions. All changes are entered directly into the manuscript in red
The abstract is very generic; the authors need to add the main results.
AUTHORS: Thank you for this observation. The main results are added in the Abstract.
The sensory analysis was directed more towards descriptive analysis and not to sensory preference tests.
AUTHORS: Thank you for this observation. After these comments, the Authors believe that we can make some changes. The authors agree with the reviewer’s comment. The performed sensory analysis was a descriptive analysis performed with trained sensory panelists. The appropriate corrections were made directly within the manuscript, firstly in the title so we changed it to “Quality and sensory profile of durum wheat pasta enriched with carrot waste encapsulates”. Due to this suggestion, we added crucial changes to the paper in order to present our results in a better manner. Thank You for this suggestion.
Line 83-85. The authors could add the concentrations of the encapsulating material with respect to the volume of waste oil extract.
AUTHORS: We measured these parameters, but we published them in our previous work (https://doi.org/10.1080/09637486.2022.2029831), so we did not show these results in this manuscript.
Line 94. Why were these FDE or SDE values chosen to replace semolina in the paste?
AUTHORS: According to the review of scientific-relevant research, we summarized that 20% replacement of semolina brings the best results, so we decided to use 10% and 20% for this investigation.
Line 97. Crude fat or lipid content?
AUTHORS: It is Crude fat (CF).
In the evaluation of optical properties (section 2.6), for total color difference (∆E*), which was the reference?
AUTHORS: Thank you for this comment. We calculated two total colour differences, one for determining encapsulate addition effect (DEAE), and second to determine the effect of pasta cooking (DECE). For the DEAE calculation we used parameters for enriched pasta samples in relation to the control pasta sample. For the DECE calculation we used parameters for uncooked pasta in relation to the cooked pasta samples. The additional comments were provided in the manuscript.
The significance letters generally reserve “letter a” for the largest mean in the tables.
AUTHORS: Thank you very much for this observation. The order of significance letters was arranged according to the Reviewer's comment.
Table 1. Why were moisture, protein and lipid contents higher when 20% BDS was used?
AUTHORS: This is explained as:
“The carrot waste encapsulates enrichment decreased moisture content and total carbohydrates, whereas increased the crude protein, crude fat, and ash. According to Gupta et al. [1], a greater protein-polysaccharides interaction in enriched samples than control leads to the reduction in the moisture content. Significantly superior protein contents were detected in pasta enriched with freeze-dried carrot waste encapsulates (FDE) which was expected due to the content of whey protein in wall material, while spray-dried encapsulates (SDE) included inulin (29%) too in wall material. The increase in protein content was 3.12 and 5.37 g/100 g with the replacement of semolina with 10% and 20% of FDE. When semolina was replaced with 10% and 20% of SDI, the increase in protein content was 1.69 and 3.85 g/100 g. Crude fat contributed to approximately 0.8% of the control pasta weight. Enriched pasta with FDE and SDE showed a great crude fat content, ranging from 4.23 to 7.20 g/100 g. This increase could be due to the inclusion of nutrients such as fatty acids present in the encapsulated carrot waste oil extract.”
Please add the units for the textural properties reported in Table 3.
AUTHORS: Thank you for this comment. The appropriate units were provided directly in Table 3.
Line 333-345. The authors need to indicate the results from Table 3 in the discussion of texture.
AUTHORS: This is indicated as:
“The texture of pasta is one of the most significant indicators of quality that influences sensory attributes and final consumer acceptance. Table 3 illustrated the textural properties of cooked pasta represented in hardness, springiness, cohesiveness, gumminess, chewiness, and resilience parameters. An increase in hardness was recorded comparing control and enriched pasta with carrot waste encapsulates. Ogawa and Adachi [27] ascribed the strength of the gluten network as the main factor that governed the hardness of enriched pasta. The samples enriched with 20% FDE and SDE showed a significant decrease in springiness and cohesiveness. This meant that for 20% enriched pasta was more difficult to hold the structure together as the coming time proceeded [28]. Gumminess and chewiness showed increasing values for FDE increasing concentration in pasta, while in terms of pasta enriched with SDE these values were lower. In addition, it was observed that the addition of carrot waste encapsulates up to 20% did not have a significant influence on pasta resilience”
Why did the authors not measure the carotenoid content in the carrot waste encapsulates and paste?
AUTHORS: We measured these parameters, but we published it in our previously work (https://doi.org/10.1080/09637486.2022.2029831) which was the first part of this investigation and development of the pasta enriched with carrot waste encapsulates.
Reviewer 2 Report
The manuscript entitled ”Quality and consumer preferences of durum wheat pasta enriched with carrot waste encapsulates” presents an interesting topic.
In general, this manuscript is well written.
The Introduction section provide background about the topic.
The experimental design is adequately discussed.
Interesting results were obtained by suitable methods and data interpretation is robust, valid and reliable.
The Conclusion section was supported by the results, was well written and provides a good conclusion for the study.
Minor comments:
In Table 3, please insert the measure units for textural parameters.
Author Response
The authors would like to thank the Editor and Reviewers for a professional review as well as the opportunity to make essential and crucial changes in our work. All Reviewer' remarks are accepted and the paper is changed according to comments. The authors believe that the changed paper would satisfy the Reviewer' criteria and that it is going to be acceptable for publishing in the Food.
We decided to revise the manuscript indicating with red letters all changes directly on the revised manuscript.
Best regards,
Dr. Olja Šovljanski and co-authors
Reviewer #2:
The manuscript entitled ”Quality and consumer preferences of durum wheat pasta enriched with carrot waste encapsulates” presents an interesting topic.
In general, this manuscript is well written.
The Introduction section provide background about the topic.
The experimental design is adequately discussed.
Interesting results were obtained by suitable methods and data interpretation is robust, valid and reliable.
The Conclusion section was supported by the results, was well written and provides a good conclusion for the study.
AUTHORS: We appreciate the time and effort that You and the reviewers dedicated to providing feedback on our manuscript and are grateful for the insightful comments and valuable improvements to our paper.
Minor comments:
In Table 3, please insert the measurement units for textural parameters.
AUTHORS: Thank you for this observation, we added the measure units in Table 3.